# Distribution and Current State of Molecular Genetic Characterization in Pathogenic Free-Living Amoebae

**DOI:** 10.3390/pathogens11101199

**Published:** 2022-10-18

**Authors:** Alejandro Otero-Ruiz, Leobardo Daniel Gonzalez-Zuñiga, Libia Zulema Rodriguez-Anaya, Luis Fernando Lares-Jiménez, Jose Reyes Gonzalez-Galaviz, Fernando Lares-Villa

**Affiliations:** 1Programa de Doctorado en Ciencias Especialidad en Biotecnología, Departamento de Biotecnología y Ciencias Alimentarias, Instituto Tecnológico de Sonora, Ciudad Obregón 85000, Sonora, Mexico; 2Programa de Maestría en Ciencias en Recursos Naturales, Departamento de Biotecnología y Ciencias Alimentarias, Instituto Tecnológico de Sonora, Ciudad Obregón 85000, Sonora, Mexico; 3CONACYT-Instituto Tecnológico de Sonora, Ciudad Obregón 85000, Sonora, Mexico; 4Departamento de Ciencias Agronómicas y Veterinarias, Instituto Tecnológico de Sonora, Ciudad Obregón 85000, Sonora, Mexico

**Keywords:** free-living amoebas, FLA, genotypes, molecular epidemiology, genomic epidemiology, *Balamuthia mandrillaris*, *Naegleria fowleri*, *Acanthamoeba* spp., *Vermamoeba vermiformis*, *Sappinia pedata*

## Abstract

Free-living amoebae (FLA) are protozoa widely distributed in the environment, found in a great diversity of terrestrial biomes. Some genera of FLA are linked to human infections. The genus *Acanthamoeba* is currently classified into 23 genotypes (T1-T23), and of these some (T1, T2, T4, T5, T10, T12, and T18) are known to be capable of causing granulomatous amoebic encephalitis (GAE) mainly in immunocompromised patients while other genotypes (T2, T3, T4, T5, T6, T10, T11, T12, and T15) cause *Acanthamoeba* keratitis mainly in otherwise healthy patients. Meanwhile, *Naegleria fowleri* is the causative agent of an acute infection called primary amoebic meningoencephalitis (PAM), while *Balamuthia mandrillaris*, like some *Acanthamoeba* genotypes, causes GAE, differing from the latter in the description of numerous cases in patients immunocompetent. Finally, other FLA related to the pathologies mentioned above have been reported; *Sappinia* sp. is responsible for one case of amoebic encephalitis; *Vermamoeba vermiformis* has been found in cases of ocular damage, and its extraordinary capacity as endocytobiont for microorganisms of public health importance such as *Legionella pneumophila, Bacillus anthracis*, and *Pseudomonas aeruginosa*, among others. This review addressed issues related to epidemiology, updating their geographic distribution and cases reported in recent years for pathogenic FLA.

## 1. Introduction

The protozoa called free-living amoebae (FLA) are dispersed in nature, with some genera with pathogenic potential for humans [1]. Currently, several species of the genus *Acanthamoeba* are known to cause infections, and only one species of each genus (*Balamuthia* (*Balamuthia mandrillaris*), *Naegleria* (*Naegleria fowleri*), *Sappinia* (*Sappinia pedata*), and *Vermamoeba* (*Vermamoeba vermiformis*)) has been described as pathogen for humans and other mammals [2,3]. Furthermore, these microorganisms except *S. pedata* are scattered on all continents. On the other hand, only *N. fowleri* infection has been related to the year’s season since these tend to increase in the summer [2,4,5]. The identification of these amoebas has changed over time. The genus *Acanthamoeba* was initially described with 20 species based on morphological criteria that included the size and shape of the cysts; however, this method has limitations because the culture conditions can modify the morphology of the cyst, and it can be variable within a single strain [6]. Therefore, molecular biology techniques have improved the identification of *Acanthamoeba* species, mainly through the analysis of the small subunit of the ribosomal RNA gene (SSU-18S-rRNA); 23 genotypes have been classified, named T1 to T23, T4 being the most frequently isolated in clinical cases due to granulomatous amoebic encephalitis (GAE) and *Acanthamoeba* keratitis (AK) [7]. Genotypes like T1, T2, T5, T10, T12, and T18 have been related to GAE, while T2, T3, T5, T6, T10, T11, T12, and T15 have been identified in AK, reporting to date around 150 cases and more than 3000, respectively [2,8,9,10,11,12]. In this context, *N. fowleri* has been characterized in 8 genotypes, of which only genotypes 1, 2, 3, 4, and 5 have been isolated in clinical samples, reporting around 381 cases of primary amoebic meningoencephalitis (PAM) [13,14,15]. On the other hand, there is little knowledge of the genetic aspects of *B. mandrillaris*, the only pathogenic species described in this genus without genotypic classification, causing around 200 infections of *B. mandrillaris*-amoeba encephalitis (BAE) and skin lesions by this protist [16,17]. An amoeba that has begun to be of great interest in the last years is *V. vermiformis*; this has been isolated coexisting in infections caused by AK and bacterial keratitis from contaminated contact lenses. Additionally, a case not related to keratitis has been described, where it generated ulceration in the patient’s eyelid [18]. Finally, there is a registered case of meningoencephalitis caused by *Sappinia pedata* (previously identified as *Sappinia diploidea*), considering it another FLA linked to infections in the central nervous system [19].

Finally, by uniting molecular epidemiology data with advances in sequencing technologies, we can say that an era of genomic epidemiology has emerged, improving traditional methods of molecular diagnosis and genotyping, even replacing them with new methods based on high-throughput genomics. Furthermore, genome sequencing has also been used to describe essential aspects of infectious diseases, such as transmission, biology, and epidemiology [20,21]. Therefore, this review’s objective is to update the geographical distribution of the FLA mentioned above, the latest infections reported by pathogenic amoebas capable of damaging the central nervous system, their genetic changes over time, and the importance and the need to exploit the enormous application of genomic epidemiology for the study of diseases caused by amoebas.

## 2. *Naegleria fowleri*

As mentioned, PAM is a disease produced by *N. fowleri*. This FLA can be found in lakes and ponds. Some of the symptoms of this infection are severe frontal headache, fever, nausea, and vomiting [22]. It is a complex disease to detect since the symptoms can be confused with other infections caused by bacteria, fungi, and viruses. Furthermore, the mortality rate is 96% because it is usually diagnosed with the disease in an advanced stage [23]. To date, eight genotypes have been characterized, seven located in Europe. They are distinguished according to the length of the internal transcript spacer 1 (ITS1) and a nucleotide transition (C/T) at position 31 of the 5.8S rRNA sequence. All genotypes are unevenly distributed in the world. Previous to this review, the most extensive study about the geographic distribution of *N. fowleri* genotypes was described by De Jonckheere [14]. However, Figure 1 shows the current geographic distribution, including those isolated from clinical cases and environmental samples [13,14,24,25,26,27,28,29,30,31,32,33].

The countries with the highest number of PAM registered cases are the United States (41%), followed by Pakistan (11%), and Mexico (9%); patients were predominantly male (75%) in the range of 1 month–85 years with a median age of 14 years being the majority due to contaminated water exposures [15]. Some of the cases reported in the last five years are shown in Table 1; the aspect being highlighted is the “time of diagnosis”, which refers to the time between hospital admission and PAM diagnosis. We use this criterion in all the tables where we mention this point. For example, a 43-year-old man diagnosed with PAM in Zhejiang, China, was recently reported with *N. fowleri* genotype 2 infection [25]. Moreover, the case of a 15-year-old young man in Nilphamari, Bangladesh, was diagnosed with PAM due to a genotype 4 strain [31]. Lastly, a 13-years-old young man in Florida, USA, was diagnosed with PAM, and the strain belonged to genotype 1 [34]. On the other hand, in years before those presented in Table 1, there was a case from Mexico of a 10-year-old boy diagnosed with PAM who had a history of swimming in an irrigation canal; the presence of *N. fowleri* in the cerebrospinal fluid was identified, and later the strain with genotype 2 characteristics was observed [14,32]. The report described above is one of the few where an early diagnosis of PAM occurs, and the patient survived the infection.

## 3. *Acanthamoeba* spp.

The *Acanthamoeba* genus was initially classified into three groups based on characteristics such as the size of the cyst and the number of arms found within it [42]. However, since these morphological characteristics change, the molecular classification in genotypes based on regions of the 18S rRNA gene is currently used [43]. Genotypes differ when there is a 5% variation between the gene’s nucleotide sequence. In addition, sequences with a <5% divergence between T2 and T6 genotypes have been determined, giving rise to the T2/T6 supertype, while the T4 has seven subtypes due to allelic diversity identified by nucleotide variation in the DF3 region of the 18S rRNA gene. These allele differences are probably indirectly related to *Acanthamoeba* species but do not represent individual species. Otherwise, more than 150 species would be described; however, allele groups can help define the limits of several evolutionary units regardless of the genus to which they correspond [44,45,46]. To date, 23 genotypes have been established: T1-T12 [43], T13 and T14 [47], T15 [48], T16 [49], T17 [50], T18 [51], T19 [52], T20 [53], T21 (WGS Project: https://www.ncbi.nlm.nih.gov/nuccore/CDEZ00000000.1/ (accessed on 18 January 2022), T22 [54], and T23 [55].

Symptoms related to infectious keratitis vary based on geographic regions, living environments, and climatic conditions. The predisposing factors to AK are corneal trauma influenced by exposure to contaminated water or soil. Interestingly, in most clinical AK cases, T4 predominates. Other genotypes associated with ocular infection include T2, T3, T5, T6, T10, T11, T12, and T15 [11,12]. In addition, the predisposing factors to contracting GAE are found in individuals with a compromised immune system such as HIV-infected patients, with presence of hematological neoplasms, with organ transplantation, who ingest steroids or follow other immunosuppressive therapy, who suffer from systemic lupus erythematosus, or with *Diabetes mellitus* or with other factors such as prolonged and excessive use of antibiotics, chronic alcoholism, liver cirrhosis, malnutrition, pregnancy, surgical trauma, burns, wounds, or radiotherapy. The T4 is commonly described in GAE cases; however, other genotypes such as T1, T2, T5, T10, T12, and T18 have been related to this disease [9]. In Table 2, the classification is broken down into the representative strains; the current distribution of genotypes and morphology worldwide of the *Acanthamoeba* spp., including those isolated from clinical cases and environmental samples, can be seen in Figure 2 [6,44,50,52,54,55,56,57,58,59,60,61,62,63,64,65,66,67,68,69,70,71,72,73,74,75,76,77,78,79,80,81,82,83,84,85]. Finally, the recent cases in databases are described in Table 3.

## 4. *Balamuthia mandrillaris*

In the genus *Balamuthia*, only two species have been described, *B. spinosa* [17] and *B. mandrillaris* [101]; the first was described at the beginning of 2022, and it is unknown if this species is pathogenic. In the same study, a 3rd species was proposed, but it is still inconclusive to date, *B. mandrillaris*. Since its isolation in 1986, approximately 200 cases have been registered worldwide; of these, 96 were in America [17,102]. In 1990 it was recognized as the causative agent of BAE, which affects the central nervous system in both immunocompromised and immunocompetent individuals. In addition, some infections can cause skin lesions [103]. The record of recent infections caused by *B. mandrillaris* is described in Table 4, and Figure 3 shows a geographical distribution of *B. mandrillaris* isolated from clinical cases and environmental samples [104,105,106,107,108,109,110,111,112,113,114,115,116,117].

*B. mandrillaris* has been isolated from environmental samples (water, soil, and dust) and cases of infections in humans and animals. Interestingly, although it is distributed among all continents, it is more common in the southern United States and South America (especially Peru) [118]. Cope et al. [119] discussed the epidemiology issue in the United States and the clinical characteristics of BAE from 1974–2016 and hypothesized that the hot and dry weather of the southwest side of the country would be the type of environment in which *B. mandrillaris* thrives. This agrees with BAE cases reported in Peru [120], which emphasize the increase in temperature as a favorable predisposing factor to contracting the infection. There are two remarkable hypotheses about the factors related to *B. mandrillaris* infections: (1) Hispanic ethnic groups comprise a higher percentage of the population in the southwestern United States, probably because of the climate of that region [119], and (2) they have frequent contact with the soil due to work activities such as farming and gardening [121]. Therefore, we can mention both the environmental characteristics where the development of this amoeba is favored and the constant exposure to soil could be related to infections by this pathogen.

Another notable *B. mandrillaris* characteristic is the difficulty of being detected due to its resemblance to histiocytes under the microscope and its unique culture requirements. Furthermore, unlike other FLA, *B. mandrillaris* is more demanding to be grown on agar because it only feeds on mammalian cells and maybe other amoebae and microorganisms or their metabolites. Moreover, healthy people can be seropositive for *Balamuthia*-antibodies due to the widespread presence of the amoeba in the environment, while people with BAE show low titers. Finally, cerebrospinal fluid analysis rarely shows the organism’s presence; for these reasons, the mortality rate is around 98%, and research on pathogenesis is challenging [104,122].

Regarding genetic characterization, an attempt has been made to genotype *B. mandrillaris* as described for *N. fowleri* and *Acanthamoeba* spp., comparing sequences of the 18S rRNA gene and the 16S rRNA mitochondrial gene. As a result, no variations were found in the 18S rRNA and low levels of variation in the 16S rRNA sequences. Furthermore, both sequences were consistent, and there is no genotype classification for this FLA [123]. Currently, there are nine complete mitochondrial genomes deposited in the National Center for Biotechnology Information (NCBI) of *B. mandrillaris* isolates (GenBank access: KT175738, KT175739, KT175740, KT175741, KP888565, KT030670, KT030671, KT030672, and KT030673), of which seven of these genomes underwent a phylogenomic analysis [124]. The phylogeny revealed the presence of at least three *B. mandrillaris* lineages, highlighting four strains of human cases from California clustered in a clade (KT030671, KT030672, KT030673, and KP888565), while in the second group, the type strain V039 together with a case of human infection was observed (KT175738 and KT175741, respectively). Finally, the third clade and the most distant phylogenomically was the V451 strain from a clinical case (KT030670). Moreover, it was determined that the variable length of the rps3 type II intron (region of the mitochondrial genome) between the different strains of this pathogen suggests that it may be an attractive target for the development of molecular genotyping. However, it is necessary to sequence more amoeba genomes from different geographical locations to respond to this hypothesis.

**Table 4 pathogens-11-01199-t004:** Cases of infections by *Balamuthia mandrillaris* reported in the last five years.

Age (Years)	Symptoms	Time of Diagnosis	Reference
13	Bitemporal abdominal pain and headache	8 days	[125]
69	Chronic nasal rash and ring lesions on the right frontal lobe	19 days	[122]
74	Drowsiness and high temperature	10 days	[126]
69	Confusion and expressive dysphasia	N/A	[102]
71	Partial spasm from the left thigh and fever	10 days	[106]
13	Fever, vomiting, headaches, and right esotropia	2 years	[115]
51	Fever, headache, altered mental status, disorientation, and seizures	3 weeks	[127]
9	Fever, headache, altered state of consciousness, and blurred vision	2 days	[16]
60	visual field loss, trouble walking, and fever	N/A	[128]
50	Headache, dizziness, dysarthria, and aphasia	N/A	[129]
17	Swelling in the left nostril, headache, weight loss, and fever	N/A	[116]
54	Numbness, weakness in left extremities	44 days	[130]
37	Dizziness	9 days	[131]
15	Fever and altered mental status	N/A	[132]
4	Epilepsy	N/A	[133]

N/A = data not available.

## 5. *Sappinia pedata*

Within the genus *Sappinia*, three species have been described: *S. diploidea* [134], *S. platani* [135], and *S. pedata* [134]. Moreover, another species is mentioned as “*S. diploidea*-like” strains [136]. In 2001, a case of human encephalitis was reported due to a FLA notably distinct to *Acanthamoeba* spp., *N. fowleri,* and *B. mandrillaris*; the infection was determined as encephalitis by *Sappinia* [137]. The diagnosed patient was a farmer suggesting the infection from inhalation of animal feces. His symptoms included vomiting, headache, loss of consciousness, and blurred vision. Fortunately, the patient fully recovered after prolonged treatment with azithromycin, intravenous pentamidine, itraconazole, and flucytosine [105,135,137].

Although *S. diploidea* was initially identified as the pathogen responsible for the case of encephalitis by morphological analysis, it was later described as *S. pedata* using two PCR assays based on 18S rRNA gene sequences. Due to the limited information on *Sappinia* spp., [138] used the 18S gene sequences to design primers and TaqMan probes that first amplified the *Sappinia* genus and later designed other primers and TaqMan probes to differentiate *S. diploidea* from *S. pedata*. The purpose was to determine if the reported case of encephalitis had been due to an *S. diploidea* infection since there was insufficient evidence supported by only analyzing images and morphology of the protozoan. As a result, the amplification of *S. pedata* from a sample of the patient’s stored tissue was observed with the qPCR assay, concluding that the pathogen that caused the disease was more related to *S. pedata* [138]. This study opened the door to the existence of other possibly undescribed species or even pathogenic free-living amoebae yet to be identified.

Finally, in the molecular genetic analyzes described, the ITS regions of four isolated strains from New Zealand, the United States, and Ukraine have been characterized, which contain the complete sequence of the ITS 1, 5.8S, and ITS 2 regions, having a size variation between 1065 to 1191 bp (GenBank accession: EU004598, EU004600, EU004597, and EU004599). Moreover, these four strains’ complete 18S rRNA gene was sequenced, and the analysis showed a size variation from 2506 to 2524 bp (GenBank accession: EU004593, EU004594, EU004595, and EU004596) [139]. However, because there is only one clinical case linked to this microorganism, there is still little information about it, both genetically and epidemiologically. For this reason, more genetic analyzes directed at other genomic regions or even other genes are required since it could be a pathogenic FLA under certain immunological circumstances of the host, or perhaps there is some environmental interaction that can give some *Sappinia* strains the pathogenic capacity. In addition, the existence of some genetic changes not yet identified is also feasible.

## 6. *Vermamoeba vermiformis*

The ability of *V. vermiformis* as a potential parasite for humans remains under discussion since the cases where it is found as the solely responsible pathogen are few. Consequently, some research has suggested that *V. vermiformis* is part of the environment microorganisms without involvement in human pathogenesis [140]. On the other hand, numerous cases have been reported where this FLA can be isolated from AK infections and bacterial keratitis. Recently, this amoeba was described as responsible for an infection in a patient’s eyelid, identified by both microbiological and molecular techniques. It is also attributed to a vector’s capacity for endocytobionts, mentioning some pathogens such as *Legionella pneumophila*, *Bacillus anthracis*, *Pseudomonas aeruginosa*, *Stenotrophomonas maltophilia*, and *Campylobacter jejuni*, each one known to be of importance for public health. Moreover, there are interactions between *V. vermiformis* with organisms of the fungi kingdom of medical importance, such as *Exophiala dermatitidis*, *Aspergillus fumigatus*, *Fusarium oxysporum*, and *Candida* spp. [18]. Furthermore, members of giant virus families were also obtained by co-culturing with *V. vermiformis*, such as Faustoviruses, which are distantly related to the mammalian African swine fever virus, Kaumoebavirus found in sewage water, and Orpheovirus IHUMI-LCC2 isolated from a rat stool sample [141].

This protozoan has been isolated and found on all continents (Figure 4) [4,18,76,142]. Although the biotic and abiotic parameters that could influence its presence in various environments have not yet been defined, it is a FLA that has caught the interest of researchers and could be considered relevant in terms of public and environmental health. However, molecular analyzes focused on this protozoan have been limited to the sequencing of the 18S rRNA gene, showing a high degree of conservation among its nucleotide sequences, suggesting that this region may be insufficient to describe the presence of differences between its virulence [4,18]. Despite this, its importance has been rooted more in its ability to vector other pathogens, as we have mentioned previously. Another proposal could be that this amoeba is opportunistic in coexisting with some set of microorganisms whose metabolites could serve as accelerators of their reproduction, or by simple co-infection, it acquires mechanisms together with other pathogens that help it enter the host.

## 7. The FLA Molecular Epidemiology and the Genetic Changes

The environment and climatic changes could affect the transmission of different pathogens and human susceptibility to them. In addition, related global warming factors, such as demographic changes and the increase in urbanization of some populations, cause the interaction of human beings with microorganisms that did not occur before [143]. The FLA infections induced by *B. mandrillaris*, *N. fowleri*, and *Acanthamoeba* spp. are attracting attention as a growing cause of parasitic death worldwide. With the climate changes mentioned above, the number of available environmental niches can be expanded and increase the frequency of “FLA-human” contact, affecting public health by interacting in its parasitic form or transmitting medically important endocytes [144,145].

Indisputably, the knowledge of genotypes in pathogenic FLA is currently relevant to recording the worldwide distribution of variants with a pathogenic history of these protozoa. For *N. fowleri*, eight genotypes are known, but only five have been isolated from clinical cases (1, 2, 3, 4, and 5). The *Acanthamoeba* genus has been divided into 23 genotypes. However, there is no sufficient evidence to classify them into virulent and non-virulent genotypes. Currently, 11 genotypes implicated in the medical field have been identified (T1, T2, T3, T4, T5, T6, T10, T11, T12, T15, and T18), highlighting the T4 genotype as the most prevalent and predominant characterized with greater virulence and less sensitivity to chemotherapeutic agents [9,11,12,146,147]. This protist is known for its extraordinary ability to interact with other microorganisms such as bacteria, fungi, viruses, and virophages, which may be one of the main reasons for a broader spectrum of genetic variability reflected in a more significant number of genotypes in comparison with others pathogenic amoebas. Another FLA related in recent years as a vector of other pathogenic microorganisms is *V. vermiformis*, which has been isolated from AK cases and other pathogens such as *L. pneumophila*, *B. anthracis*, and *P. aeruginosa*, among many others. Contrary to *Acanthamoeba* spp., *V. vermiformis* does not undergo lateral gene transfer processes, at least not enough to acquire genotypic variability. However, studies of this amoeba are not as extensive as those that cause GAE and PAM.

Regarding *B. mandrillaris*, genotyping is also not applied; although isolates have been recovered from cases in humans, mandrill, and environmental samples such as soil and water, there are still no marked genetic changes in this genus. Finally, after more than 30 years, the second *Balamuthia* species was recently reported (*B. spinosa*) by Lotonin [17], and the debate on the possibility of a third has been opened. Although the pathogenicity in the new species is not known, it is interesting how, with the development of technology, new species have been described within genera that have been unique for many years. Expanding the knowledge about these FLA is essential; describing the distribution of these protozoans worldwide is crucial to keep genetic analyses and molecular epidemiology at the forefront. Furthermore, since climate change and the expansion of urbanization are inevitable, more variations could occur over time, and these reports need constant updating.

## 8. Advances in Genomic Epidemiology

There are advances in epidemiology migrating from molecular epidemiology to genomic epidemiology thanks to the development of genomics, the increase in computational resources, and the design of pipelines to analyze metadata quickly and efficiently. Genomic epidemiology seeks to link knowledge of pathogen genomes with information on disease transmission, monitoring factors related to outbreaks, and creating tracking networks that aim to discover the spatial scales of transmission, and demographics that contribute to transmission patterns and forecast epidemic tendencies [148]. For example, it is expected that, as a result of the pandemic caused by SARS-CoV-2, the epidemiology approach to analyzing structural variants of the viral genome promoted the development of research of this nature in the process of describing the reasons for the existence of variants with greater virulence and reported the changes that this pathogen acquired over time in different geographical locations, predicting a greater probability of increased contagion and the characteristics of the infection depending on the main variant in circulation among the population [148,149,150,151,152]. On the other hand, genomic epidemiology is also applied to understand other types of diseases, mainly those caused by *Escherichia coli* [153,154], *Shigella* spp. [155], *Klebsiella pneumoniae* [156], *Corynebacterium diphtheriae* [157], *Staphylococcus aureus* [154,158], *Enterococcus faecalis* [154], and *Mycobacterium tuberculosis* [159]. Although this discipline is more developed in bacteria and now in viruses, we can find extensive reports on emerging fungal pathogens [160,161,162,163,164] describing antifungal resistance mechanisms or shared synteny among geographically diverse species. Finally, the pathogens we are interested in discussing are protozoa, specifically the FLA. Therefore, we can start by indicating that these infections are the least studied at the level of genomic epidemiology; however, we describe the cases where essential findings have been reported through genomic approaches.

Liechti et al. [165] performed a comparative genomic analysis of two non-pathogenic species of *Naegleria* genus (*N. gruberi* and *N. lovaniensis*) with *N. fowleri*. A close relationship between *N. lovaniensis* and *N. fowleri* was confirmed by protein clustering methods. The high similarity and close relationship of these two amoebas mentioned above provide the basis for new comparative approaches at the molecular and genomic level to discover pathogenicity factors in *N. fowleri*. Subsequently, an omics approach analysis was carried out to understand the biology and infection of *N. fowleri* by sequencing new strains as well as using transcriptomic analysis of low and high virulence cultured in a mouse infection model and comparing it with *N. gruberi*. The analysis concluded by observing high gene expression levels for various enzymes previously considered pathogenicity factors in *N. fowleri.* The principal genes were actin, the prosaposin precursor gene of *Naegleria* pore A and B, phospholipases, and Nf314 (Cathepsin A); they also identified key individual targets, such as the S81 protease and two cathepsin B proteases, which are missing in *N. gruberi* and differentially expressed in *N. fowleri* passaged by mouse [166]. The need for the availability of multiple high-quality genomes from other non-pathogenic species is notable for improving comparative genomics evaluation.

Moreover, a comparative genomics study of seven *Acanthamoeba* spp. clinically pathogenic strains identified possible genes specific to this genus related to virulence. The authors reported genes coding for metalloprotease, laminin-binding protein, and heat shock proteins (HSP) and identified several endosymbionts such as species of *Chlamydiae*, *Mycobacterium*, *Legionella*, *Burkholderia*, *Rickettsia,* and *Pseudomonas*, which were significantly different among the isolated, being *Pseudomonas* spp. the most common bacteria shared in all strains, so they discussed the possibility that this genus may be closely related to virulence and pathogenicity; however, more studies are needed to prove this hypothesis [167].

There is currently no comparative genomic analysis of *B. mandrillaris*. Nevertheless, metagenomic analysis has been used to diagnose this pathogenic protozoan in samples from sick patients; but since it is a disease with a high mortality rate, unfortunately, most of them did not survive [16,115,128,168]. In addition, the development of new compounds against *B. mandrillaris* has been limited by the scarcity of genomic information, so a recent study performed a transcriptome analysis with the identification of 17 target genes, and 15 were validated by direct sequencing; these genes would be helpful to develop new approaches for the treatment of *B. mandrillaris* infections [169]. For this pathogen, in particular, it is necessary to expand the information on the omics technologies because it can remain in the host for years [131], in addition to presenting characteristics such as difficulty in its identification employing conventional techniques (molecular, microbiological, or histopathological) and the disease presenting different times of expression.

The infections caused by the FLA described above represent a significant challenge that requires specific public health strategies, especially in low-income countries, but unfortunately, many of them are considered neglected diseases whose main consequences include affecting physical and cognitive development; limiting individual productivity; generating economic problems [170]; and finally, causing the death of children, young people, and adults indiscriminately. Undoubtedly, the need to create epidemiological programs to understand pathologies and the spread of diseases such as GAE, PAM, AK, and skin lesions denotes a broad panorama to be explored through structural and functional genomics, supported by bioinformatic tools that allow the creation of algorithms for the analysis of sequences in mass and the creation of an epidemiological network directed to protozoa. In summary, taking advantage of the benefits that technology offers us by applying them to aspects like (1) gene-environment interactions, (2) factors that may contribute to the development of free-living amoebas or other protozoa diseases in populations (systems epidemiology), (3) identification of genetic variants related to changes in infections and the severity of the disease, (4) description of new genotypes and emerging virulence phenotypes in response to environmental changes, and of course, (5) design of new diagnostic methods and epidemiological monitoring.

## Figures and Tables

**Figure 1 pathogens-11-01199-f001:**
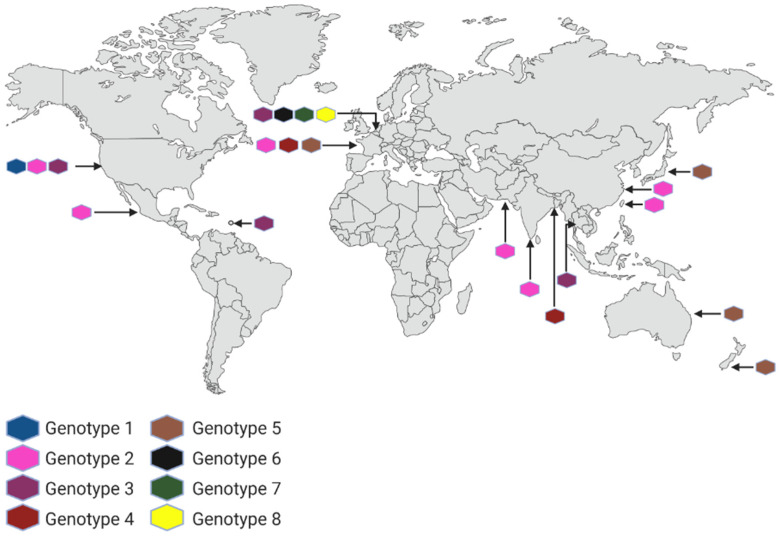
Geographical distribution of genotypes (clinical and environmental cases) of *N. fowleri* worldwide. Created with BioRender.com (accessed on 26 August 2022).

**Figure 2 pathogens-11-01199-f002:**
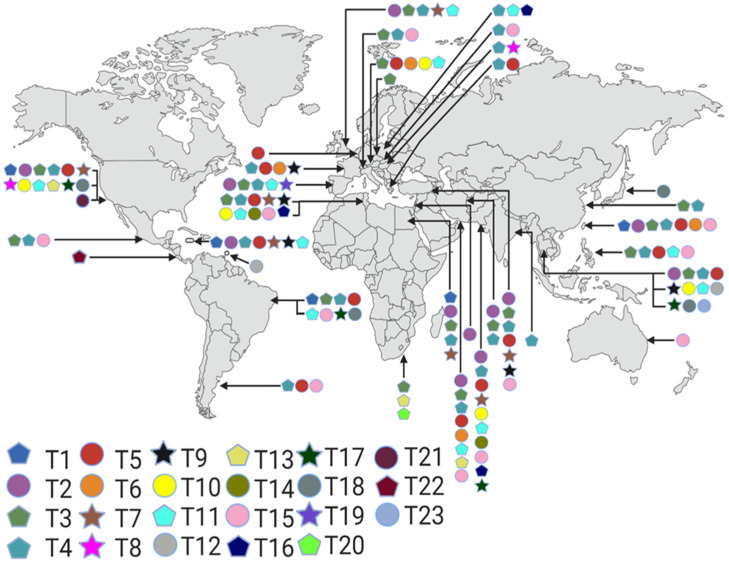
Geographical distribution of genotypes (T1-T23) and morphology (clinical and environmental cases) of *Acanthamoeba* spp. Star, pentagon, and circle correspond to groups 1, 2, and 3 according to the morphological relationship of the cysts. Created with BioRender.com (accessed on 11 July 2022).

**Figure 3 pathogens-11-01199-f003:**
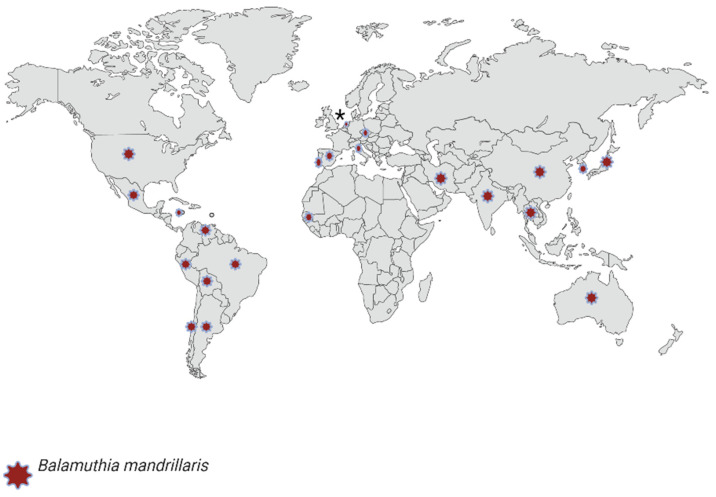
Geographic distribution of isolates (clinical and environmental cases) of *B. mandrillaris* globally. Created with BioRender.com (accessed on 3 October 2022). ***** Study presenting the clinical case in the Netherlands is inconclusive as to whether the infection was acquired in Gambia or the Netherlands.

**Figure 4 pathogens-11-01199-f004:**
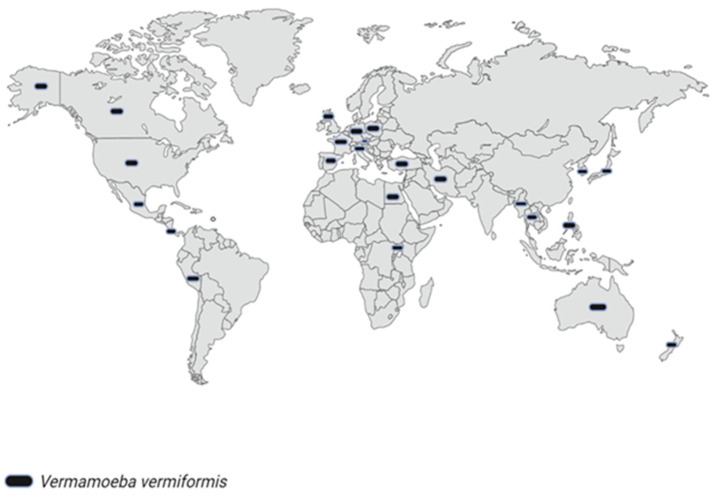
Geographical distribution of *V. vermiformis* in the world, mostly from isolates in cases of keratitis caused by other organisms. Created with BioRender.com (accessed on 26 August 2022).

**Table 1 pathogens-11-01199-t001:** The last five years of PAM cases around the world.

Location	Age (Years)	Symptoms	Genotypes	Time of Diagnosis	Reference
China	42	Headache and dyspnea	N/A	2 days	[35]
China	43	Headache, fever, fatigue, pain, and chills	2	N/A	[25]
Bangladesh	15	Fever, severe headache, vomiting, weakness, and stiff neck	4	3 years *	[31]
Zhejiang	43	Headache, fever, myalgia, and fatigue	N/A	4 days	[36]
USA	8	Headache, stiff neck, photophobia, and vomiting	N/A	2 days	[37]
Costa Rica	15	General discomfort, intense headaches, nausea, and vomiting	N/A	7 days	[38]
Costa Rica	5	Headaches, vomiting, fever, lower limb pain, spasticity, hyperreflexia, and walking difficulty	N/A	N/A	[38]
Costa Rica	1	Fever, drowsiness, and an altered state of consciousness	N/A	4 days	[38]
Pakistan	26	High fever, vomiting, and altered consciousness	N/A	4 days	[39]
China	8	Headache, fever, vomiting, seizures, and coma	N/A	24 days *	[40]
USA	13	Headache, vomiting, and fever	1	N/A	[34]
Turkey	18	Fever, nausea, vomiting, and changes in consciousness	N/A	7–10 days *	[41]

* postmortem; N/A = data not available.

**Table 2 pathogens-11-01199-t002:** Classification of genotypes and representative subtypes of the genus *Acanthamoeba*.

Genotype	Strain	GenBank Accession Number
T1	*A. castellanii* V006	U07400
T2	*A. palestinensis* Reich	U07411
T2/6A	*A. polyphaga* CCAP 1501/3b	AY026244
T2/6B	Isolate OB3b_3A	AB425945
T2/6C	*A. palestinensis* OX-1	AF019051
T3	*A. griffini* S-7	U07412
T4A	*A. castellanii*	U07413
T4B	*A. castellanii* Ma	U07414
T4C	*A.* sp. Fernandez	U07409
T4D	*A. rhysodes* Singh	AY351644
T4E	*A. polyphaga* page-23	AF019061
T4F	*A. triangularis* SH621	AF346662
T4 Neff	*A. castellanii* Neff	U07416
T5	*A. lenticulata* Jc-1	U94739
T6	*A. palestinensis* 2802	AF019063
T7	*A. astronyxis*	AF019064
T8	*A. tubiashi* OC-15C	AF019065
T9	*A. comandoni*	AF019066
T10	*A. culbertsoni* Lilly A-1	AF019067
T11	*A. hatchetti* BH-2	AF019068
T12	*A. healyi* V013 OC-3A/AC-020	AF019070
T13	UWC9	AF132134
T14	PN13	AF333609
T15	*A. jacobsi* 31-B	AY262360
T16	U/HC1	AY026245
T17	Ac E1a	GU808277
T18	CDC:V621	KC822461
T19	USP-AWW-A68	KJ413084
T20	OSU 04-020	DQ451161
T21	*A. royreba*	CDEZ01000000
T22	*A. pyriformis*	KX840327
T23	*A. bangkokensis*	MZ272148 and MZ272149

**Table 3 pathogens-11-01199-t003:** Reported cases of *Acanthamoeba* infections from different genotypes in the last five years.

Sampling Date	Age (Years)	Symptoms	Genotype	Time of Diagnosis	Reference
2018	29	Sensitivity to light, redness, and pain in the right eye	N/A	1 day	[86]
2018	27	Severe eye pain, blurred vision, photophobia, and a foreign body sensation in the left eye	T4	1 day	[87]
2018	34	Redness of the left eye and blurred vision	N/A	6 months	[88]
2018	40	Corneal ulcer and infected left eye	T3	6 months	[84]
2018	69	Central corneal ulcer deepening with hypopyon	T4	>8 months	[84]
2018	41	Central corneal abscess with hypopyon	T4	5 months	[84]
2018	22	Corneal ulcer in the right eye	T4	7 months	[84]
2018	28	Red eye and foreign body sensation for a week	T4	1 week	[89]
2018	54	Progressive neurological deficits, including sensorimotor paralysis of the right leg and impaired alertness	T4	12 days	[90]
2019	38	Photophobia and inflammation	T4	8 months	[91]
2019	24	Multifocal stromal infiltrates	T4	3 weeks	[62]
2019	28	Multifocal stromal infiltrates	T8	2 weeks	[62]
2019	53	Altered mental status and fever	N/A	2.5 months	[92]
2019	9	Decreased vision, pain, photophobia, and redness	T15	~5 weeks	[93]
2020	65	Bilateral *Acanthamoeba* panophthalmitis (first case)	N/A	N/A	[94]
2020	54	Blurred vision, photophobia, and excessive lacrimation	N/A	2 weeks	[95]
2020	18	Eye pain and blurred vision associated with redness, swelling, and eye discharge	N/A	7 weeks	[95]
2021	30	Irritation, itching, pain, and redness in the right eye	N/A	5 days	[96]
2021	59	Redness, tearing, pain, and white discoloration of the cornea in the right eye	N/A	~2.5 months	[97]
2021	45	Redness, pain, and tearing in the right eye	N/A	N/A	[98]
2022	27	Fever, chills, and lethargy	T1	N/A	[99]
2022	24	Decreased vision, severe pain, eyelid swelling, redness, and irritation in the right eye	T4	N/A	[100]
2022	23	Pain, hypopyon, and irritation in the left eye	T4	N/A	[100]
2022	42	Vision loss, photophobia, and irritation in both eyes	T4	N/A	[100]

N/A = data not available.

## Data Availability

Not applicable.

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
