# Peer review of "Distribution and Current State of Molecular Genetic Characterization in Pathogenic Free-Living Amoebae"

_pathogens, 2022, doi:10.3390/pathogens11101199_

Round 1

Reviewer 1 Report

This manuscript reviews the geographic distribution of FLA and their genetic characterization. It highlights that most FLA infections are limited to the warmer regions and that our understanding of these infections is rather limited. This is a helpful review and for the most part well written and researched. There are however instances where citations have been made incorrectly, and too many references are made to review articles instead of the primary literature.  Minor points are raised below.

The sentence “However, few genera of FLA are linked to human infections. Within these genera, Acanthamoeba spp., classified by genotypes (T1-T23), being T1, T2, T4, T5, T10, T12, and T18 as capable of causing granulomatous amoebic encephalitis (GAE) in immunocompromised patients mostly and Acanthamoeba keratitis related to genotypes T2, T3, T4, T5, T6, T10, T11, 22 T12 and T15 in apparently healthy patients.” Is awkward and should be rewritten. I suggest, “Some genera of FLA are linked to human infections. The genus Acanthamoeba is currently classified into 23 genotypes (T1-T23), and of these some (T1, T2, T4, T5, T10, T12, and T18) are known to be capable of causing granulomatous amoebic encephalitis (GAE) mainly in immunocompromised patients while other genotypes (T2, T3, T4, T5, T6, T10, T11, T12 and T15) cause Acanthamoeba keratitis mainly in otherwise healthy patients.

Line 27. I suggest Sappinia sp. as the species that caused this single case has only been indirectly determined and it is now clear that there are many more species in this genus than had been previously supposed (also in line 40, 65 and thereafter).

Line 51 “being the T4 the most isolated” better as “T4 being the most frequently isolated” and also this observation needs to be supported by a reference.

Line 80 A reference is needed for the symptoms of PAM especially the contention that dysgeusia is a symptom?

Line 91. PAM not MAP

Line109. Table 1. It is not clear what “Time of diagnosis” means? Is this after the exposure event or is this the time between symptoms and diagnosis, or between admission and diagnosis? Also, the second case NA should be N/A like the others

Line 117.  Better as “since these morphological characteristics change”.

Line 118. Reference 41 is not correct this should be the Stothard reference currently 45

Line 163. References are required for these two species

Line 180. Perhaps mention that the higher incidence of Balamuthia in Hispanic people is probably because these people are more likely to be exposed to soil rather than to infer that they are genetically more susceptible as there is no data to support the late?

Line 211 Figure 3. I believe that the single Balamuthia case reported from the UK was that of a motorcyclist who contracted his infection in Peru? If this is correct its probably more correct to attribute that infection to Peru rather than the UK?

Line 220 Table 4 the “time of diagnosis” needs to be explained as for table 1

Line 226. This is incorrect. S. pedata was not identified on the basis of sequence as this was not determined. This was based on primers producing a PCR product only and is unsafe.

Line 247. The term “environmental pollutant” is not appropriate here, this amoeba is merely part of the normal microfauna.

Author Response

This manuscript reviews the geographic distribution of FLA and their genetic characterization. It highlights that most FLA infections are limited to the warmer regions and that our understanding of these infections is rather limited. This is a helpful review and for the most part well written and researched. There are however instances where citations have been made incorrectly, and too many references are made to review articles instead of the primary literature.  Minor points are raised below.

  • Dear Reviewer, we appreciate your positive comments on our manuscript. It is enriching for us to know your opinion and suggestions, which I explain below how we resolved them.

The sentence "However, few genera of FLA are linked to human infections. Within these genera, Acanthamoeba spp., classified by genotypes (T1-T23), being T1, T2, T4, T5, T10, T12, and T18 as capable of causing granulomatous amoebic encephalitis (GAE) in immunocompromised patients mostly and Acanthamoeba keratitis related to genotypes T2, T3, T4, T5, T6, T10, T11, 22 T12 and T15 in apparently healthy patients." Is awkward and should be rewritten. I suggest, "Some genera of FLA are linked to human infections. The genus Acanthamoeba is currently classified into 23 genotypes (T1-T23), and of these some (T1, T2, T4, T5, T10, T12, and T18) are known to be capable of causing granulomatous amoebic encephalitis (GAE) mainly in immunocompromised patients while other genotypes (T2, T3, T4, T5, T6, T10, T11, T12 and T15) cause Acanthamoeba keratitis mainly in otherwise healthy patients.

  • Thank you very much for the suggestion. We have changed the sentence, and it reads better that way.

Line 27. I suggest Sappinia sp. as the species that caused this single case has only been indirectly determined and it is now clear that there are many more species in this genus than had been previously supposed (also in line 40, 65 and thereafter).

  • We have read more closely and agree to make the change because some strains are listed as " diploidea-like" (Corsaro et al., 2017) or that the case of encephalitis was "probably" caused by S. pedata (Qvarnstrom et al., 2009).

Line 51 "being the T4 the most isolated" better as "T4 being the most frequently isolated" and also this observation needs to be supported by a reference.

  • Thanks again for the suggestion. We have changed the redaction and added the reference where the T4 genotype is described as the most frequent.

Line 80 A reference is needed for the symptoms of PAM especially the contention that dysgeusia is a symptom?

  • In this case, the symptoms caused by PAM had been described as mentioned by Myint et al., (2012)*. However, it is not described as the primary symptom by the CDC (https://www.cdc.gov/parasites/naegleria/illness.html). Therefore, although It is probable that there will be disorders in the sense of smell and taste (https://www.cdc.gov/parasites/naegleria/clinical-features.html), since it is not one of the main symptoms, we have eliminated it.

*Myint, T., Ribes, J. A., & Stadler, L. P. (2012). Primary amebic meningoencephalitis. Clinical Infectious Diseases, 55(12), 1737-1738.

Line 91. PAM not MAP

  • An apology for the error; we have made the relevant change.

Line109. Table 1. It is not clear what "Time of diagnosis" means? Is this after the exposure event or is this the time between symptoms and diagnosis, or between admission and diagnosis? Also, the second case NA should be N/A like the others

  • We have explained the meaning in lines 95 – 97 and clarified that this criterion will be used in the tables where the "time of diagnosis" is mentioned. We also corrected the NA in Table 1. Thank you very much for the comment.

Line 117.  Better as "since these morphological characteristics change".

  • You are correct. We have made the suggested change.

Line 118. Reference 41 is not correct this should be the Stothard reference currently 45.

  • I apologize for the confusion. We have added Stothard et al., (1998) in that section.

Line 163. References are required for these two species

  • References have been added on lines 173 – 174

Line 180. Perhaps mention that the higher incidence of Balamuthia in Hispanic people is probably because these people are more likely to be exposed to soil rather than to infer that they are genetically more susceptible as there is no data to support the late?

  • Sorry for the confusion. We didn't want to link genetic predisposition to infections but to the environment where amoeba is more likely to develop. So in lines 186 -196, we describe the hypotheses about the infection factors and add soil exposure as one of them for better clarity of our conclusion on this point according to what we have read in the literature.

Line 211 Figure 3. I believe that the single Balamuthia case reported from the UK was that of a motorcyclist who contracted his infection in Peru? If this is correct its probably more correct to attribute that infection to Peru rather than the UK?

  • Due to your comment, we re-revised the publication of this report and have corrected the image. However, the case was assigned to Bolivia since Saffioti et al., (2022)* mentioned the case described by White et al., (2004)**, who identified that geographical point as the place where the patient traveled (motorcyclist).

*Saffioti, C.; Mesini, A.; Caorsi, R.; Severino, M.; Gattorno, M.; Castagnola, E. Balamuthia Mandrillaris Infection: Report of 1st Autochthonous, Fatal Case in Italy. European Journal of Clinical Microbiology and Infectious Diseases 2022, 41, 685–687, doi:10.1007/s10096-022-04404-9.

**White JML, Barker RD, Salisbury JR et al (2004) Granulomatous amoebic encephalitis. Lancet 364(9429):220. https://doi.org/10.1016/ S0140- 6736(04) 16640-3

Line 220 Table 4 the "time of diagnosis" needs to be explained as for table 1

  • We explain it in lines 95 and 97 so that it is not repetitive in each table.

Line 226. This is incorrect. S. pedata was not identified on the basis of sequence as this was not determined. This was based on primers producing a PCR product only and is unsafe.

  • Dear Reviewer, this comment led us to carefully read again the methodology used by the authors to identify pedata in their analyses. *Qvarnstrom et al., (2009) performed both types of PCR; firstly, PCR to amplify the 18S gene from different Sappinia samples, and later they sequenced the PCR products. Based on these sequences, they designed primers and TaqMan probes to perform qPCR assays. The first set of primers + probes was to identify the genus Sappinia; once this objective was achieved, they used strains of S. pedata, S. diploidea, and other FLA (Acanthamoeba, Naegleria, and Balamuthia) and a sample of the stored tissue of the patient reported in 2001 with encephalitis caused by Sappinia. When performing the qPCR, it can be seen that the amplification corresponding to the tissue sample is specific for S. pedata. Although we agree that more studies are needed on Sappinia species, the study described employed qPCR in this case. We add this explanation to the manuscript for greater clarity.

*Qvarnstrom, Y., Da Silva, A. J., Schuster, F. L., Gelman, B. B., & Visvesvara, G. S. (2009). Molecular confirmation of Sappinia pedata as a causative agent of amoebic encephalitis. The Journal of infectious diseases, 199(8), 1139-1142.

Line 247. The term "environmental pollutant" is not appropriate here, this amoeba is merely part of the normal microfauna.

  • Thanks for the comment. We have modified the term (line 275-276).

Reviewer 2 Report

- line 91: please change MAP with PAM.

- line 151: in Table 2 there is the classification of Acanthamoeba genotypes, however only T1-T14 genotypes are reported. I recommend adding the others too, from T15 to T23.

- lines 163-165: I recommend to add a citation on this affirmation.

- lines 223-224: I recommend to add a citation on this affirmation.

- line 263: please correct "an FLA" with "a FLA"

- line 304: please modify MAP with PAM.

Author Response

Reviewer 2.

  • Dear Reviewer, thank you very much for your kindness and recommendations; we have changed the text as requested.

- line 91: please change MAP with PAM.

  • Thanks for the correction; we have changed MAP to PAM on line 91

- line 151: in Table 2 there is the classification of Acanthamoeba genotypes, however only T1-T14 genotypes are reported. I recommend adding the others too, from T15 to T23.

  • You are right; sorry for the mistake. We have placed the correct table with the 23 genotypes.

- lines 163-165: I recommend to add a citation on this affirmation.

  • Thanks for the suggestion. We have added the corresponding references on lines 173 – 174

- lines 223-224: I recommend to add a citation on this affirmation.

  • Gracias por la sugerencia, hemos agregados las citas correspondientes en la líneas 238 - 240

- line 263: please correct "an FLA" with "a FLA"

  • Again, thanks for the comment. We have modified it to the correct form you request

- line 304: please modify MAP with PAM.

  • Thanks for the correction. We have changed MAP to PAM on line 332.
